



# Climatic and tectonic controls on shallow marine and freshwater diatomite deposition through the Palaeogene

Cécile Figus[1,2], Or M. Bialik[3,4], Andrey Yu. Gladenkov[5], Tatyana V. Oreshkina[5], Johan Renaudie[6], Pavel Smirnov[7], and Jakub Witkowski[1]

[1]Institute of Marine and Environmental Sciences, University of Szczecin, Szczecin, 70-383, Poland

[2]Doctoral School of the University of Szczecin, Szczecin, 70-383, Poland

[3]Institute of Geology and Paleontology, University of Münster, Münster, 48149, Germany

[4]Dr. Moses Strauss Department of Marine Geosciences, The Leon H. Charney School of Marine Sciences, University of Haifa, Carmel, 31905, Israel

[5]Geological Institute of Russian Academy of Sciences, Moscow, 119017, Russia

[6]Museum für Naturkunde, Berlin, 10115, Germany

[7]Petroleum Higher School, Almetyevsk, 423450, Russia

**Correspondence:** Cécile Figus (cecile.figus@phd.usz.edu.pl)

**Abstract.** Diatoms play a major role in the carbon and silicon cycles, and thus diatom-bearing sediments represent an archive of past climatic and environmental settings. In shallow marine and freshwater environments, the accumulation of diatom frustules forms a sedimentary rock called diatomite. However, most global scale studies of diatom-bearing sediments focus on deep-sea sites, whereas shallow marine and freshwater diatomites are studied mainly at a regional level. To address this

problem, we present a global scale compilation of diatomite occurrences spanning the Palaeogene (~66 to ~23 Ma). This period was characterized by initial extreme warmth followed by a prolonged cooling, disrupted by short-term climatic events called hyperthermals, and by a number of palaeoceanographic and palaeogeographic changes. The aim of this compilation is to determine the response of diatom production to Palaeogene environmental fluctuations, by examining the influence of climate, tectonic activity and ocean circulation on diatomite deposition. Although climatic factors appear to have had an indirect impact,

our study suggests that palaeogeographic and palaeoceanographic changes were key drivers of diatomite deposition during the Palaeogene, particularly from the Early Eocene Climatic Optimum (~53 to ~49 Ma) onwards. In fact, our compilation suggests the absence of diatomite deposition in epicontinental seas between ~46 and ~43 Ma, while diatomites do not begin to accumulate in open ocean environments until ~43.5 Ma. Moreover, we observe that regional climate and volcano–tectonic activity have had an impact on the deposition of freshwater diatomites.

# 1 Introduction

Diatoms are currently the main primary producers and exporters of organic carbon and silica in the oceans (Barron et al., 2015; Renaudie, 2016). These unicellular microalgae are photosynthesic, which enables atmospheric $CO_2$ to be transferred to the oceans, resulting in the fixation of organic carbon and the release of $O_2$ (Guidry et al., 2007; Penman et al., 2020). Diatoms absorb the silicic acid ($H_4SiO_4$) available in the ocean's surface waters to develop hydrated silica exoskeletons, called frus-



tules (Kooistra et al., 2007), which are then either dissolved or deposited on the ocean floor and buried in sediments (Yool and Tyrrell, 2003; Guidry et al., 2007; Katz et al., 2024). These processes occur in shallow and deep marine settings, as well as in freshwater environments on continents. Sediments containing diatoms, therefore, constitute archives of past geological and environmental processes. The ability of certain assemblages of diatom taxa to adapt to different climatic and environmental conditions provides information about the environments in which they developed (Kooistra et al., 2007). Consequently, di-

atoms are a well-suited proxy for palaeoenvironmental and palaeoceanographic reconstructions. However, our understanding of diatom responses to past palaeoceanographic events is mostly shaped by data from the deep ocean. Numerous deep-sea drilling campains (e.g. Deep Sea Drilling Project - DSDP, Ocean Drilling Program - ODP, and International Ocean Drilling Program - IODP) have been generating pelagic diatom records from the global ocean, whereas studies of onshore sites are generally at local or regional scales (e.g., McLean and Barron, 1988; Dunbar et al., 1990; Strelnikova, 1992; Radionova et al.,

1999, 2003; Kvaček, 2002), and few studies link sites from different regions (Barron et al., 2015; Witkowski et al., 2020). One possible explanation may be found in the scarcity of active fossil marine diatoms workers (Witkowski, 2018). Another may be the vulnerability of diatoms to diagenetic processes (Knoll et al., 2007), resulting in discontinuities in the record that increase with sediment age.

    While the presence of diatoms since the Cretaceous is well documented, the date of their first appearance in the Mesozoic

remains uncertain (Bryłka et al., 2023) and their importance in the Palaeogene period is not completely understood. The Palaeogene (~66 to ~23 Ma) is divided into three epochs, beginning with a particularly warm climate during the Palaeocene–early Eocene, slowly cooling towards the Oligocene (Cramwinckel et al., 2018). Multiple short-term climate disturbances, often of extreme magnitude, occurred during the early Palaeogene. These so-called 'hyperthermal' events are characterized by a rapid (between 1,000 and 100,000 years) increase in temperature, lasting from 100,000 years to 2 million years, coupled with a rise in

atmospheric $CO_2$, leading to a negative carbon excursion (CIE) and ocean acidification (Foster et al., 2018). The modification of the carbon cycle, and subsequently the silicon cycle, through the intensification of continental weathering correlated with warm temperatures and elevated atmospheric $CO_2$ (Berner et al., 1983; Penman et al., 2019), led to increased availability of silicic acid in the oceans for silica-secreting biota. Diatoms would, therefore, be more likely to form extensive blooms during hyperthermal events, but Van Cappellen et al. (2002) suggest that over short periods, corresponding to the residence time of

silica in the oceans, the input of aluminum from silicate weathering may offset the effects of increased $[H_4SiO_4]$ on marine biosiliceous production by reducing the efficiency of silica recycling in the oceans. However, the quantity of living diatoms in the oceans (Leblanc et al., 2012) is such that the Si/C ratio in the biogenic pool (Brzezinski, 1985) should be about 10 times higher than the silica flux brought to the oceans by rivers (Tréguer et al., 1995). Thus, there is a hysteresis in the system, due to the storage and recycling of silica in the biogenic pool, which could be as much as 10 times longer than the residence time

calculated by simply dividing the fluxes by the concentrations. Furthermore, the inclusion of radiolarians and fossil diatoms in the calculation could actually indicate a longer residence time, since they can potentially increase the total biogenic pool. This hypothesis suggests that after the end of the hyperthermal events, there would still be time to use the additional silica.





The aim of the present work is to verify the response of diatom production to changes in the Palaeogene climate, using diatomite records from freshwater and shallow marine environments. The frequency of diatom-bearing sediments in individual
stratigraphic intervals is expected to reveal the response of diatom production to climate change.

To this end, we present a compilation of Palaeogene shallow marine and freshwater diatomite occurrences, gathering records on a global scale, with age control relying mainly on diatom biostratigraphy. Based on these data, and using reconstructed sea levels and carbon and oxygen stable isotope records, we examine the co-occurence of peaks in diatomite frquency with the Palaeocene-Eocene Thermal Maximum (PETM; ~56 Ma) and the Early Eocene Climatic Optimum (EECO; ~53 to ~49 Ma).
We also provide new insights into (1) the role of diatoms in palaeoceanographic and climatic fluctuations, and (2) the depositional drivers of diatomite in shallow marine and freshwater environments during the Palaeogene. Moreover, the compilation and analysis of these data have enabled the detection of a previously unknown gap in the deposition of shallow marine diatomite between ~46 and ~43 Ma, which does not correspond to any climatic event documented thus far.

## 2 Materials and Methods

### 65 2.1 Lithology

Diatoms develop in aquatic environments, under the control of physico–chemical factors such as light and nutrient availability (Battarbee et al., 2001), affecting the type of sediment formed by the accumulation of diatom frustules.

This study is based on a compilation of geographic and stratigraphic diatomite occurrences, including altered diatomite. According to the nomenclature established by Zahajská et al. (2020), diatomite is a consolidated sediment with a high porosity
(> 70%) and contains more than 50% diatoms by weight of sediment. This includes diatomites whose matrix is diluted by other types of sediment, such as clay or sand, hence the names sandy or clayey diatomites, which are also considered in the compilation.

Diagenetic processes have a major impact on diatomite, for example transforming opal-A through an increase in temperature and pressure with growing burial depths, and can lead to the formation of sedimentary rocks such as chert or porcelanite
(Zahajská et al., 2020). Therefore, freshwater and shallow marine sedimentary rocks containing diatoms—and specifically recognized in the literature as originating from diatomite—are included in the dataset.

Diatomaceous sediments and sedimentary rocks are not included in this compilation, due to the level of uncertainty associated with the characterization of these sediments in the literature.

### 2.2 Geographic distribution

The compilation, based on a literature survey involving different types of publications (i.e. geological reports, articles on taxonomy and biostratigraphy), covers 102 diatomite occurrences from the Palaeogene (Fig. 1). The dataset is divided into 12 geographic regions: Transuralia/West Siberia, Kazakhstan, Dnieper-Donets Basin, Middle Volga Basin, Kamchatka, western Europe, northern Europe, eastern Europe, North Africa, North America, South America and Oceania.



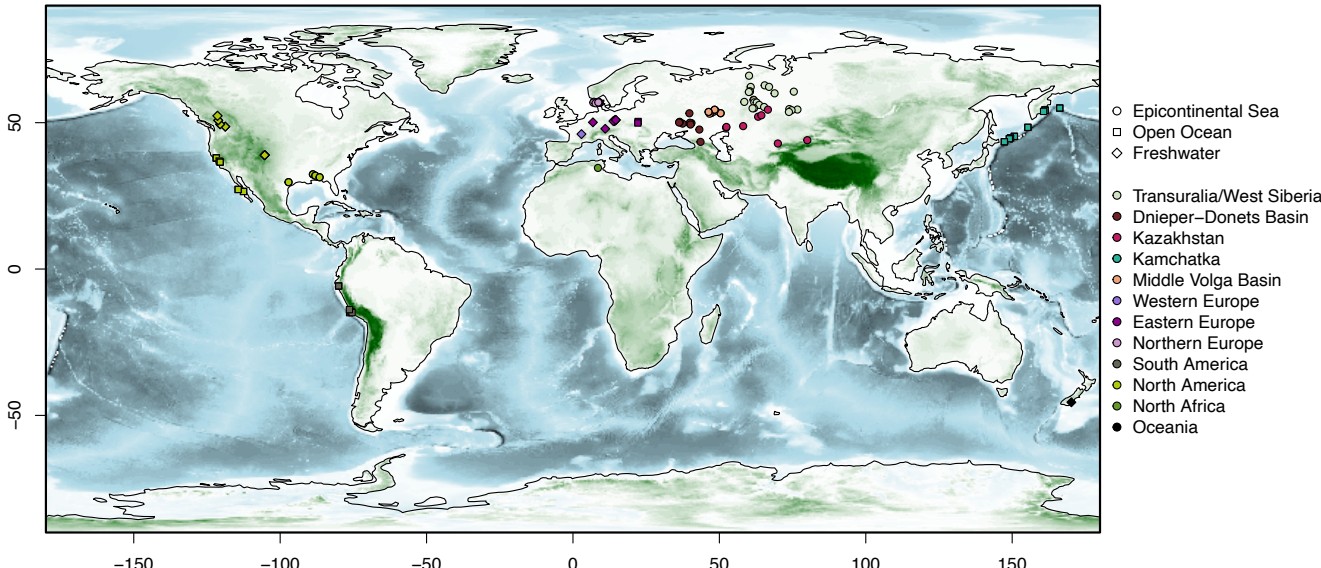

**Figure 1.** Map showing all the sites compiled in this study, with the colours corresponding to the 12 basins/regions and the shapes corresponding to the type of environment (epicontinental sea, open ocean or freshwater; see legend). Bathymetry from ETOPO 2022 (NOAA NCEI, 2022).

Site selection is based on the availability and accuracy of the following parameters: age, geographic coordinates, and lithol-
ogy. No restrictions are applied on the location of the sites, in order to create a global-scale compilation. However, sites without
a precise location are rejected so as to not lose track of the characteristics of their depositional environments. It should be noted
that the number of sites for each region is strongly related to the available literature (i.e. the large number of well-documented
occurrences on the Eurasian Plate). Despite careful research, other diatomite occurrences may not be represented, due to the
lack of available information (e.g. in the southern hemisphere), or due to linguistic issues. As the literature survey relies heavily
on online searches, papers not indexed in English, French, Spanish, German, Polish or Russian (the languages that the authors
use), could potentially escape our attention.

A differentiation between depositional environments is also made, to enable a better understanding and interpretation of
the results. Three depositional environments are distinguished in the compilation: epicontinental seas (abbreviated as ES),
open ocean settings (OOS) and freshwater. Epicontinental seas are flooded continental areas, whether continental shelves or
larger inland parts flooded by rising sea levels, while open ocean settings refer to the deposition of diatoms on the continental
slope. Although still part of the continental crust, the continental slope is subject to open ocean conditions, which differs from
epicontinental seas in terms of ocean circulation. Finally, the freshwater environment brings together lacustrine records in
this compilation. While the separation of freshwater and shallow marine records is an obvious choice, due to the differences
between these depositional systems, a second distinction between diatomites formed in ES or OOS is necessary, in order to
follow the effects of ocean circulation on diatomite deposition.



## 2.3 Chronology

The age range in this study extends from 66 to 23 Ma, covering the Palaeocene, Eocene and Oligocene epochs.

In most cases of freshwater diatomites, a published radiometric age measured on adjacent volcanic layers allows a precise dating of lake filling. However, most of the occurrences in this compilation were not precisely dated and required biostrati-
graphic study to establish the ages. The correlation between diatomites from the Eurasian Plate and other sites can sometimes be a source of biostratigraphic uncertainty, due to differences in taxonomic assemblages or the lack of calcareous microfossils that would enable direct correlation. The biostratigraphic ages in this study are based on the direct correlation of diatoms with dinocyst zones and other microplanktonic groups, calibrated with nannofossil zonations and paleogeomagnetic timescales (Gradstein et al., 2004, 2012), and compared with diatom assemblages present in the sediments (Scherer et al., 2007).
All ages in this compilation are binned into intervals with a 1 Myr resolution and given on the Gradstein et al. (2012) timescale. Chronostratigraphic terms used throughout this paper are also consistent with Gradstein et al. (2012).

## 2.4 Statistical treatment

The tabulated number of diatomite occurrences is divided by depositional setting in Fig. 2, and by basins/geographical clusters in Fig. 3.
In order to assess the error margin induced by stratigraphic uncertainty, we attempted to measure a confidence interval on the observed temporal patterns (see Fig. 2), using a bootstrapping approach (Efron, 1992). For shallow marine sites, in each trial, half of the sites picked randomly have the upper and lower limit of their age interval modified by a randomly selected error on a normal distribution, whose standard deviation corresponds to half the length of the biozone to which the age of the sites has been assigned (for epicontinental sea: calcareous nannofossils zones; for open ocean setting: calcareous nannofossils,
radiolarians zones or magnetochrons). For freshwater sites, the standard deviation corresponds to the error margin of the radiometric estimated age. The resulting tabulations are computed over 10,000 of such trials, and the mean 95th percentile is kept as the 95% confidence interval.

The distribution of diatomites is projected with GPlates (Müller et al., 2018) onto the palaeobathymetric maps of Straume et al. (2024), using Torsvik et al. (2019) rotation model, for each 1 Myr time bin (see Supplementary Material). Four maps are
presented here for relevant time intervals: 30 Ma, 35 Ma (Fig. 4), 44 Ma and 56 Ma (Fig. 5). From these palaeobathymetric maps, the total area represented by the flooded continental platform is computed for each 1 Myr time interval (Fig. 2C)—to simplify the area where the palaeobathymetry is between 0 m and 1,000 m—in order to test whether the observed temporal pattern in shallow marine diatomite occurrences is primarily an artefact of fluctuations in the available depositional area. Similarly, the temporal distribution of diatomites is compared with variations in sea level (Miller et al., 2020, ; Fig. 2D). In
both cases, these correlations, or lack thereof, are tested using a detrended linear regression analysis (Wooldridge, 1999). As the temporal resolution of the sea level curve is higher than that of the diatomite occurrences, the diatomite records are compared not only with the mean sea level during each 1 Myr time interval, but also with the minimum and maximum sea level during



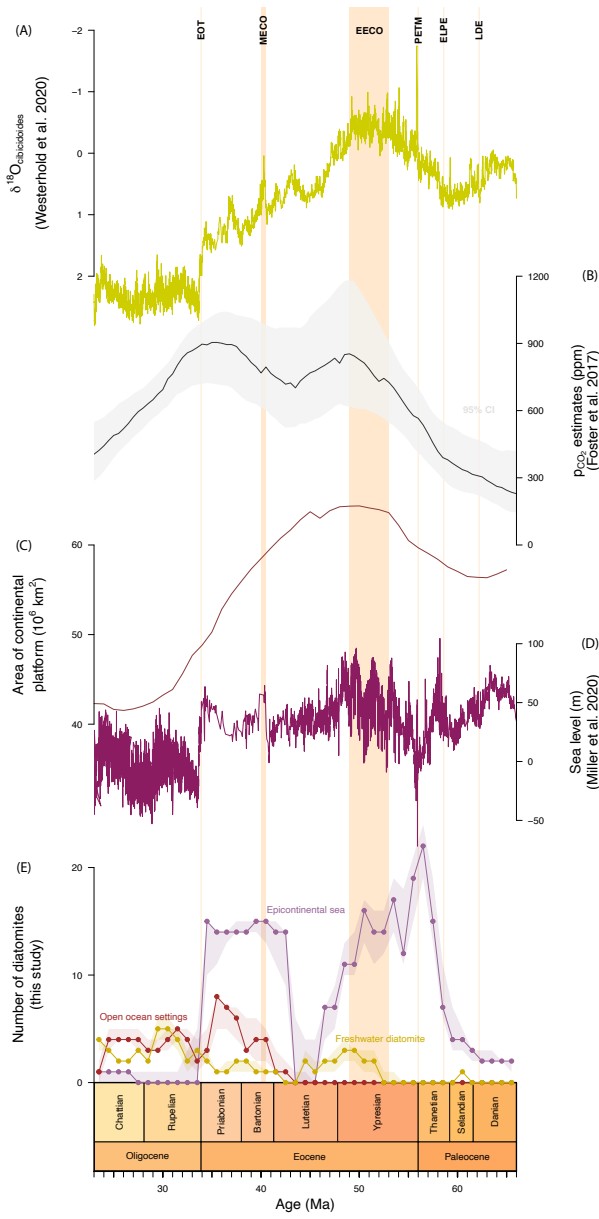

**Figure 2.** Comparisons between tabulated diatomite occurrences and various proxies. (A) $\delta^{18}O_{Cibicidoides}$ from Westerhold et al. (2020); (B) $pCO_2$ estimates from Foster et al. (2018), with mean in black and 95% confidence interval in grey; (C) estimated area of continental platform based on Straume et al. (2024) palaeotopography (see section 2.4); (D) sea level estimates from Miller et al. (2020); (E) tabulated occurrences of diatomites by environment type, with 95% confidence interval based on bootstrapped stratigraphic ages (see section 2.4 for explanations).



these intervals. In addition, the significance of the points of inflection in the diatomite occurrences discussed in this study is checked with the Chow test (Chow, 1960).

## 3    Results and Interpretations

The compilation of Palaeogene diatomite occurrences is presented in Fig. 2E, where deposits from ES, OOS and freshwater environments are plotted as separate curves. Freshwater diatomites do not seem to show any trend over the Palaeogene, whereas ES diatomite occurrences peak in the late Palaeocene and display high abundance through the early Eocene. An abrupt drop in the number of ES occurrences is observed between ~46 and ~43 Ma, and no OOS occurrences are observed prior to ~43.5 Ma.

### 3.1    Diatom production during the PETM and EECO

Hyperthermal events trigger a negative climatic feedback. Following the significant release of carbon into the atmosphere, the intensification of the hydrological cycle, elevated temperatures, and greenhouse gas concentrations, lead to an acceleration of silicate weathering on land, consuming atmospheric $CO_2$ and subsequently cooling the climate (Walker et al., 1981; Berner et al., 1983; Penman et al., 2019). Accelerated silicate weathering is postulated to cause a drastic increase in silicic acid ($H_4SiO_4$) input to rivers and, hence, increased availability of $[H_4SiO_4]$ in the oceans during the hyperthermal events (e.g., PETM; Penman, 2016). Yet, the response of diatoms to Palaeogene hyperthermal events is not completely understood. Penman (2016) proposes that elevated $[H_4SiO_4]$ in the oceans during hyperthermals may have enhanced the marine biosiliceous production, thus impacting organic carbon sequestration.

Figure 2E shows an abrupt increase in the number of ES diatomite occurrences at the end of the Palaeocene, around the Palaeocene-Eocene Thermal Maximum (PETM; ~56 Ma), then decreasing slowly towards the middle Lutetian (~45.4 Ma), apart from two peaks during the Early Eocene Climatic Optimum (EECO; ~53 to ~49 Ma). The Middle Eocene Climatic Optimum (MECO; ~40.5 to ~40 Ma) is also marked by a slightly higher number of ES diatomite occurrences, before an abrupt drop at the Eocene-Oligocene Transition (EOT; ~33.9 Ma). These results suggest a causal link between hyperthermal events and the deposition of diatomite in ES conditions. The resolution of the dataset is too coarse to determine whether peaks in diatomite deposition are associated with any specific hyperthermal events, especially during the EECO, when hyperthermals were especially numerous. Notably, however, the ES diatomite record shows stepwise increases in the number of occurrences that could be associated with the Latest Danian Event (LDE; ~62.2 Ma) and with the Early-Late Palaeocene Event (ELPE; ~58.6 Ma).

The area of continental platform (Fig. 2C) —which represents the surface area of the continental shelf flooded during the Palaeogene— expands during the Palaeocene, in keeping with reconstructed $p_{CO_2}$ levels (Fig. 2B). These results, together with the reduction in sea level (Fig. 2D) during the LDE and ELPE, may explain the trends in diatomite occurrences during these hyperthermals by a reduced availability of flooded continental platform surface (fewer epicontinental seas) and a weaker negative feedback than during the Eocene hyperthermal events. Nevertheless, detrended regression analysis indicates that area of continental platform (R-squared= 0.1612, p-value= 0.004), sea level (mean sea level: R-squared= 0.0426, p-value= 0.098;



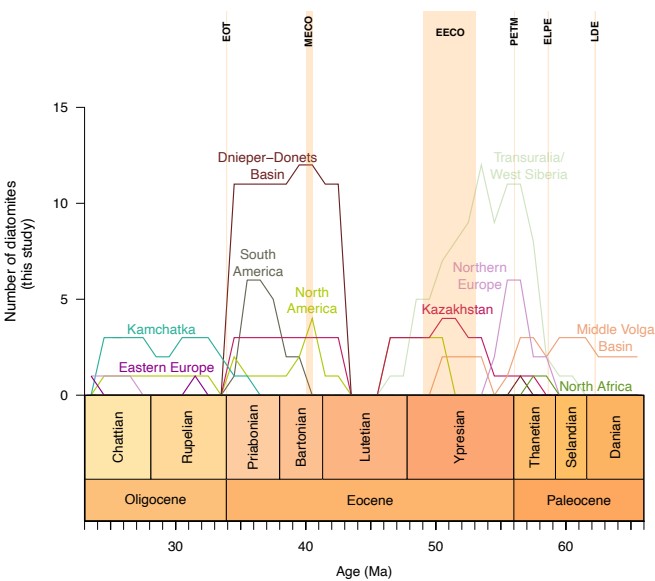

**Figure 3.** Tabulated diatomite occurrences by basin/region. Only diatomites formed in epicontinental seas and open ocean settings are shown.

minimum sea level: R-squared$= 0.0004$, p-value$= 0.319$; maximum: R-squared$= 0.024$, p-value$= 0.159$) and ES diatomite occurrences are not directly correlated throughout the record, in particular between ~46 and ~43 Ma, implying that the number of ES diatomite occurrences is not purely an artefact of the space available for deposition. Similar results can be observed in the Oligocene, where $\delta^{18}O$ (Fig. 2A), $p_{CO_2}$, area of continental platform and sea level all decrease after the EOT, as does the number of ES diatomite deposits.

On the contrary, during periods of high temperature (as suggested by $\delta^{18}O$) and high $p_{CO_2}$, ES diatomite deposition appears to rise, corroborating the link between warm temperatures, high greenhouse gas levels and enhanced marine production via the increased availability of $H_4SiO_4$.

No occurrences of OOS diatomite are found in the compilation before ~43.5 Ma, when the number of occurrences starts to increase towards the MECO (Fig. 2E). Three peaks are then present at ~37.5–35.5 Ma, ~31.5 Ma and ~27.5–24.5 Ma. These

four depositional events in the OOS diatomites seem to coincide with episodes of sea level rise (Fig. 2D). Furthermore, the presence of peaks between hyperthermal events is consistent with the upwelling-related origin of several OOS deposits. Indeed, upwelling is reduced during hyperthermal events, due to changes in temperature gradients (Griffith et al., 2021).

Finally, the freshwater diatomites in Fig. 2E do not illustrate any specific response to climate proxies, unlike the ES and OOS deposits. Therefore, several questions arise: why do ES diatomites seem to respond to the PETM and EECO, and why is

there a break in the record between ~46 and ~43 Ma? Why do the OOS diatomites in our compilation only start to accumulate from ~43.5 Ma? And lastly, what drives the deposition of freshwater diatomites?





## 3.2 Early–middle Eocene diatomites from epicontinental seas

The diatomite record in epicontinental seas implies a response of diatomite accumulation to long term climate fluctuations.
However, a drop in the number of occurrences, unrelated to any known climatic or palaeoceanographic event, is present between
~46 and ~43 Ma. Nevertheless, from the PETM onwards, the record of ES diatomite occurrences, punctuated by two peaks
spanning the EECO, decreases towards the middle Lutetian. In Fig. 3, diatomite occurrences from Transuralia and West Siberia
peak at ~53.5 Ma, while an increase occurs in Kazakhstan and the Middle Volga Basin, coinciding with the onset of the EECO.
Consequently, we suggest that the inflection point delimiting the start of the disappearance of diatomites is placed at the
beginning of the EECO (Chow test: $F = 33.6$, p-value$= 5.22 \times 10^{-7}$). The end of the event seems easier to place, with the
abrupt resumption of diatomite accumulation at ~43 Ma.

### 3.2.1 Climatic impact

A comparison between ES deposits and climate proxies between the end of the EECO and ~43 Ma suggests a correlation
between the records (Fig. 2). During the EECO, the short decline between the two peaks in ES diatomites corresponds to a rapid
drop in $p_{\mathrm{CO_2}}$ and temperature, before increasing again. After the second peak during the EECO, the number of ES diatomites
decreases slowly in a stepwise fashion, along with $p_{\mathrm{CO_2}}$ and temperature, until a drop corresponding to the disappearance
of the diatomites. From ~43.5 Ma, ES diatomite occurrences and $p_{\mathrm{CO_2}}$ begin to increase again, while temperatures reach a
peak. These results are consistent with the scenario described by Penman (2016) and highlight the indirect correlation between
climate and diatom productivity. Indeed, the ES diatomite record may illustrate long-term cooling caused by terrestrial silicate
weathering from the Eocene onwards (Walker et al., 1981; Wallmann, 2001; Torfstein et al., 2010). However, this scenario is
not sufficient to explain the cessation of diatomite deposition during the middle Eocene.

### 3.2.2 Tectonic drivers

Global palaeogeography changed markedly through the Palaeogene (Figs 4, 5, see Supplementary Material): for example, the
opening of the Tasman and Drake passages in the Southern Ocean (Vahlenkamp et al., 2018; Barron et al., 2015), the collision
of the Indian plate with the Asian continent (Dupont-Nivet et al., 2010), or the closure of the Arctic Basin connection with
Volga regions, Trans-Urals and West Siberia (Radionova et al., 2003; Smirnov and Konstantinov, 2017), and the contraction of
the Tethys Ocean (Straume et al., 2024). An examination of the lithology of the deposits present in the compilation provides
a better understanding of the events that led to the gradual disappearance of diatomites and the abrupt resumption of diatom
accumulation.

A notable feature of the Earth's continents during the late Cretaceous–early Palaeogene period is the dominance of relatively
flat landforms (Froelich, 2014; Tsekhovsky, 2015). At the Cretaceous–Palaeogene boundary, peneplain weathering crusts are
associated with destructive tectogenesis in Eurasia (Nikonova and Khudyakov, 1982; Nikonova, 1987; Milanovskii, 1995;
Tsekhovsky, 2015), which potentially provides a substantial amount of dissolved silica to regions undergoing continental or
marine sedimentation (Akhmetiev et al., 2012; Smirnov and Konstantinov, 2017; Amon, 2018). Rising concentrations of dis-






**Figure 4.** Palaeogeographic and palaeobathymetric maps representing all sites at 30 and 35 Ma. The unfilled circles are sites that did not contain diatomites during these periods, while the magenta circles are shallow marine diatomites and yellow circles are freshwater diatomites. Palaeobathymetry after Straume et al. (2024).

solved silica in river waters lead to a corresponding stimulation of siliceous biota in river–marine mixing zones. On the Eurasian

Plate (Trans-Urals, West Siberia, Kazakhstan, Dnieper-Donets Basin and Middle Volga Basin), seven levels of biosiliceous sedimentation (diatomite depositional events - DDE) have been identified (Jousé, 1978; Strelnikova, 1992; Radionova et al.,





**Figure 5.** Palaeogeographic and palaeobathymetric maps representing all sites at 44 and 56 Ma. See Figure 4 for legends.

1999, 2003; Aleksandrova et al., 2011; Olshtynska, 2013; Olshtynska and Tsoy, 2018; Oreshkina and Aleksandrova, 2017; Oreshkina et al., 2021). The tectonic structure, orography and landform characteristics of the West Siberian epicontinental basin should be identified as a key factor influencing diatomite deposition in this region. The expansion of marine areas in Western Siberia during the Palaeocene allowed extensive meridional communication between the Central Asian regions and the Arctic–North Sea region, followed by rapid closure and decline of the basin. At the Palaeocene–Eocene boundary, the North





Atlantic Igneous Provinces (NAIP) produced intense volcanic activity (Storey et al., 2007), corresponding to a peak in diatomite deposition in northern Europe (Fig. 3). The large silica influx produced by NAIP might be interpreted as a causal mechanism for diatomite deposition in northern Europe, as well as a possible impact on diatomite deposition in West Siberia (Smirnov et al., 2020). The depth of the basin along the Ural Mountains, combined with minimal detrital influx in the Palaeocene and abundant nutrients in the area, led to the formation of a considerably thick photosynthetic zone and enabled diatoms to bloom. However, in the interval from ~46 to ~43 Ma, changes in sedimentation are observed in the marginal parts of the Trans-Urals, West Siberia and Kazakhstan. The Russkaya Polyana layers (Akhmetiev et al., 2010) represent a marker of the transition from biosiliceous to terrigenous sedimentation, which inhibits the formation of diatomite (Smirnov and Konstantinov, 2017; Amon, 2018). They also represent the termination of communication between the West Siberian Sea Strait and the Arctic basin, reorganizing inter-basin circulation. Since the Eocene, the surface area of marine epicontinental basins has decreased (Fig. 2C), as has the area of distribution (Figs 4, 5, and Supplementary Materials) and thickness of the diatomites in their marginal parts. For example, the similarity of diatom assemblages found in the Middle Volga Basin and West Siberia may indicate a connection between the basins in the Palaeocene. The high marine biosiliceous production gives way to a continental environment in the Eocene, due to the tectonic reorganization closing the inter-basin connections (Radionova et al., 1999). In contrast to the Middle Volga Basin, the drastic change in palaeogeography in the Dnieper-Donets Basin during the Lutetian (Radionova et al., 1999) allows a shift in diatomite accumulation from Transuralia/West Siberia to the Dnieper-Donets Basin (Fig. 3).

The palaeogeographic reorganization of the area during the Palaeogene also affected the southern Tethys margin, with the Gafsa Basin. Kocsis et al. (2014b) suggest that the connection with the Tethys Ocean might have increased biosiliceous production via nutrient inputs from upwelling, and that Palaeocene-Eocene lithologies in the Gafsa Basin are the result of palaeogeographic reorganizations and sea level changes (Fig. 2D). In North America, the Gulf of Mexico is bordered by the U.S. Gulf Coastal Plain, in which are located Eocene ES diatomites (Fig. 1). During the Cenozoic, this region has been affected by several events of transgressive floods (Galloway et al., 2000), leading to the deposition of diatomites. In the Gulf of Mexico, diatomites and volcanic ash layers originate from volcanism and uplift in central Mexico and the southwestern United States, causing flooding of the western basin between the late Eocene and early Oligocene (Davis et al., 2016). According to Weaver and Wise (1974), U.S. Gulf Coastal Plain diatomites might be the result of an enhanced nutrient supply to the region, in conjunction with favourable ocean currents. Witkowski et al. (2021) highlight the impact of the North Atlantic circulation system on the North American coast and phytoplankton production. Nevertheless, it is interesting to note that an elevated $_{bio}SiO_2$ flux is recorded in the Blake Nose (North Atlantic) between ~46 and ~42 Ma (Witkowski et al., 2021), which corresponds to the period without diatomite occurrences in our compilation (Fig. 3). Comparison of the Blake Nose results with ours suggests a contrasting pattern between the pelagic and the nearshore realms. In our compilation, tectonic and ocean circulation play a major role in shallow marine diatomite deposition, which is expected to be different in deep-sea environments. The distribution and deposition of deep-sea diatom-bearing sediments will be examined in a future article and compared with the results of this study.



### 3.3 Diatomite deposition in open ocean settings

#### 3.3.1 Lithology

According to our study (Fig. 2E), the accumulation of diatom frustules giving rise to diatomite is estimated to have begun at ~43.5 Ma on the west coast of North and South America, in Kamchatka and in the Polish Outer Carpathians.

If we examine the stratigraphy of sites in western South America just before 43.5 Ma, it appears that the middle Eocene was eroded in several locations. Diatomites formed on the Peruvian coast, due to Eocene to Mio-Pliocene subsidence of the region initiated by subduction and erosion of the area (Malinverno et al., 2021; Di Celma et al., 2022). Thus, a major hiatus extends from the Upper Cretaceous to the middle Eocene, caused by magmatic and metamorphic activities in the arc, as well as by erosion in the region (Dunbar et al., 1990). Malinverno et al. (2021) indicate that the diatom blooms were related to seasonally limited upwelling, and suggest that their origin may be linked to the Antarctic Circumpolar Current and the initiation of a proto-Humboldt current. Furthermore, John A. Barron (personal communication, 8 May 2024) reports that *Rhizosolenia oligocaenica* Schrader (1976) and other Antarctic diatoms are present in the Otuma Formation (Peruvian coastline) of late Eocene–early Oligocene age, as well as in the lower Oligocene of Bering Island in the Komandorsky Islands at the westernmost end of the Aleutians, and in Kamchatka (Gladenkov, 2008, 2019). The Kuril-Kamchatka island arc, including the submarine Vityaz Ridge on the island slope, contains Oligocene diatomites deposited on a Palaeogene volcanogenic–sedimentary complex (i.e. conglomerates, sandstones, ignimbrites). During the Oligocene, tuffaceous diatomites and effusive volcanites were formed under subaerial conditions and at shallow depths (Lelikov and Emelyanova, 2011). In Kronotskii Bay, adjacent to the Kuril-Kamchatka Trench, Gladenkov (2008) identified Oligocene tuffaceous diatomites correlatives with the *Rocella gelida* Zone. Tsoy (2002) indicates that assemblages from the *Rocella gelida* Zone in the island slope of the Kuril-Kamchatka Trench may have formed in deeper water than assemblages from the *Rhizosolenia oligocaenica* Zone, due to the deepening of the basin during the Oligocene. In addition, Eocene diatom assemblages similar to those on the west coast of North America were discovered along volcanic and sedimentary deposits in the Ol'ga and Zhupanovskii canyons of Kronotskii Bay (Tsoy, 2003).

On the west coast of North America, Milam and Ingle (1982) suggest that the Palaeogene diatomites of California are indicative of upper and middle bathyal environments with minimal oxygen layers, probably coincident with a period of sea level rise or a tectonic event. According to Barron et al. (1984), the presence of *Fenneria brachiata* (Brightwell) Witkowski (2018) in the Kellogg Shale could represent cold-water deposition conditions, similar to those of the California Current system. Further south, diatomites from Baja California Sur reflect intense upwelling during the Eocene and possibly a tectonic movement (McLean and Barron, 1988).

Finally, in the Polish Outer Carpathians, diatomites accumulated during the Oligocene in depressions located on the basin slope (Kotlarczyk and Kaczmarska, 1987). According to Van Couvering et al. (1981), the Palaeocene–Eocene deposits of the Carpathians are typical of the deep-sea, with distal turbidites containing ichnofossils underlying the diatomaceous sediments. Oligocene diatom assemblages suggest basin subsidence, but n palaeoclimatic interpretations is provided (Kotlarczyk and Kaczmarska, 1987).





The OOS records suggest that the absence of diatomite occurrences prior to ~43.5 Ma is related to tectonic factors. It is impossible to state with certainty that no OOS diatomite accumulated before ~43.5 Ma, due to lack of published information.

However, the OOS sites gathered in this study seem to have reached optimal conditions for diatom deposition via tectonic movements, and the presence of Antarctic diatom assemblages in temperate regions of South America could indicate that ocean circulation played a major role in the accumulation of OOS diatomites during the Palaeogene.

### 3.3.2 Impact of the ocean circulation

The classic view (Haq, 1981; Keller, 1983; Barron, 1987; Miller, 1992; Huber and Sloan, 2001) of the pre-Oligocene ocean

describes it as strongly stratified and sluggish, relative to the modern ocean. Such a configuration would limit the recycling and rejuvenation of nutrients critical to phytoplankton blooms. Under those settings, terrestrially sourced nutrients would be more readily available than upwelled ones, as evidenced by the preference for nearshore environments in the early Eocene (Fig. 2E). As global cooling set in during the aftermath of the EECO, this stratification began to disintegrate with increased deep water formation in high latitudes (Lunt et al., 2017; Zhang et al., 2020). The combined effect of destratification and increased

latitudinal gradients allowed the establishment of more upwelling zones.

With intensified winds, a shift occurred from the early to the late Eocene: upwelling cells moved mainly from low latitudes (Slansky, 1980; Barusseau and Giresse, 1987; Lu et al., 1995; Charisi and Schmitz, 1998-02; Johnson et al., 2000; Soudry et al., 2006; Prian, 2014; Kocsis et al., 2014a; Meilijson et al., 2023) to mid (Slansky, 1980; Barusseau and Giresse, 1987; Johnson et al., 2000; Prian, 2014; Kocsis et al., 2014a; Wade et al., 2020) and high latitudes (Marty et al., 1988; Diester-Haass and Zahn, 2001; Diekmann et al., 2004). This increase in upwelling activity would allow more nutrient rejuvenation further from shore and an increase in OOS diatomite accumulation towards the end of the Eocene (Fig. 2E).

### 3.4 Freshwater diatomite

Twenty-one palaeolakes, from New Zealand, North America and Europe, contain Palaeogene diatomites (Figs 4, 5). According to the geology of the sites, all these lakes are associated with tectonic and/or volcanic formations (i.e. lakes in active grabens,

maar lakes).

The freshwater diatomites of the České Středohoří Mountains in northern Bohemia date from the late Eocene to the late Pliocene. Lacustrine sedimentation began with the rifting of the Palaeogene peneplain, initiated by volcanic activity in the region (Prokop, 2003). In North America, the early to middle Eocene southern Canadian Cordillera was affected by the creation of grabens and half-grabens, and by the activity of a volcanic arc producing volcanic fill in the basins. Diatom blooms

developed between eruptive episodes, leading to the formation of the diatomite occurrences recorded in this study (Mustoe, 2005, 2011, 2015-07-06), such as at Florissant (Colorado), where diatomites formed in a lake interpreted as the result of the damming of a fluvial drainage by a volcanic formation (Benson, 2011). Four maar lakes are also present in the compilation: Eckfeld Maar and Rott in Germany, Menat in France and Foulden Maar in New Zealand.

While Benson et al. (2012) argue that diatom diversity in these lakes was impacted by changes in global climate (i.e.,

hyperthermals), this does not seem to have affected the number of diatomite occurrences gathered in this study (Fig. 2E).

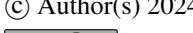

Seasonality did, though, play an important role for these diatomites, as did volcano–tectonic activity. In fact, Mustoe (2005) points out that freshwater diatom deposits are often linked to volcanism, due to the large silica input provided by weathering of volcanic glass. The early Eocene Giraffe Pipe maar lake is a good example, with its uniquely preserved diatom-bearing sediments, illustrating biodiversity shifts in relation to changes in the environment and climate (Siver and Lott, 2023).

**4   Conclusions**

The number of shallow marine diatomite occurrences appears to be indirectly correlated with Palaeogene climate change, but the resolution of the data does not allow the establishment of a precise link between diatomite deposition and individual hyperthermal events. Palaeogeographic reorganization and ocean circulation are identified as the main drivers of changes in diatomite deposition during the Palaeogene, and explain the absence of diatomite occurrences in our compilation between 330 ~46 and ~43 Ma. In addition, comparison of this study with published results from the Blake Nose (North Atlantic) suggests differences in the deposition of diatom-bearing sediments between shallow and deep marine environments, which will be discussed in a future article.

Although freshwater diatomites do not respond to climatic and palaeogeographic fluctuations on the same spatio-temporal scale as shallow marine diatomites, our study highlights that seasonality and volcano-tectonic activity have a major impact on 335 lacustrine diatoms.

*Data availability.*   The diatomite compilation can be found on Zenodo (Figus et al., 2024). Palaeogeographic maps showing all diatomites in each 1 Myr time bin can also be found in the Supplementary Materials.

*Author contributions.*   JW designed the study. CF, AYG, TVO and JW prepared the compilation. CF, AYG and TVO carried out biostratigraphic dating. CF and JR processed the data. All co-authors participated in the interpretation of the data. CF prepared the manuscript, with 340 contributions from all co-authors.

*Competing interests.*   The authors declare that they have no conflict of interest.

*Acknowledgements.*   This study was co-funded by the Polish Minister of Science as part of the "Regional Initiative of Excellence" programme for 2024-2027 (RID/SP/0045/2024/01). We would like to thank John A. Barron for his help in determining the depositional environments of the sites on the Pacific coast of North and South America.



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
