# Peer review of "Climatic and tectonic controls on shallow marine and freshwater diatomite deposition through the Palaeogene"

_EGUsphere, 2024_

## Referee Comment (RC1)

**Review of "Climatic and tectonic controls on shallow marine and freshwater diatomite deposition through the Palaeogene" by Cécile Figus, Or M. Bialik, Andrey Yu. Gladenkov, Tatyana V. Oreshkina, Johan Renaudie, Pavel Smirnov, and Jakub Witkowski. (EGUSphere in review). August 23, 2024.**

**Reed Scherer, Northern Illinois University**

**General Comments:**

Diatoms play a key role in global biogeochemical cycling, contributing to atmospheric oxygen levels and to CO2 regulation through sequestration in geological deposits. Diatoms require high nutrients to bloom and accumulate, so intervals of diatom proliferation on and near continental masses typically correlate with times of rapid continental weathering and delivery to non-marine and marginal marine waterways; the latter including both epicontinental seas and along continental shelves, especially near river mouths that carry a high dissolved nutrient load but low sediment load. By contrast, deep-sea diatomite accumulation is typically dominated or augmented by upwelling of nutrients at oceanic divergence zones, driven by broad ocean circulation patterns. These may be along continental margins (e.g, California coast), or in open ocean settings (e.g., the Southern Ocean).

Figus et al. have analyzed occurrences of Paleogene diatomites around the world, based on literature surveys, seeking patterns that may relate to global climate changes in the Hothouse world. The analysis they present is based on literature surveys of terrestrial exposures. Their paper is an ambitious compilation that provides an important window into a critical interval in Cenozoic history. Their chief aim is to evaluate the hypothesis that Paleogene diatomite accumulation was driven predominantly by climate change. One notable new observation is a possible "diatomite gap" between ~46 and ~43 Ma.

There are, of course, biases and limitations of the stratigraphic record, which they acknowledge. The greatest limitation is access to material for study. On the continents, marine diatomites will only accumulate during times and in places affected by of high sea level. Access these strata is limited because they may have become deeply buried with little or no exposure, or, once exposed, these deposits are highly susceptible to erosion. Consequently, terrestrially-based Paleogene diatomites are very rare. Access to appropriate deep-sea deposits is even more challenging. Most known oceanic Paleogene diatomites were recovered by ocean drilling around the world and are not part of this paper. Some Paleogene open ocean diatomaceous deposits are found on land in some coastal areas, having been tectonically uplifted. Paleogene deep-sea sediments, accessible only through ocean drilling, may be deeply buried, increasing their susceptibility to diagenesis and also making them less accessible through high quality drilling. Moreover, Paleogene diatomites that would have accumulated in the deep sea across global oceans have

already been subducted and destroyed, thus very severely limiting assessment of global Paleogene deep sea accumulation.

**Specific Comments:**

The authors limit their analysis to sediments both exposed on land and technically defined as diatomite, excluding diatom-bearing muds. They distinguish three categories of Paleogene diatomite: freshwater diatomites, epicontinental sea deposits (ES) and open ocean settings (OOS). Their database lists 102 terrestrially exposed diatomites used in the analysis. Of these only 17 are classified as OOS and 20 as freshwater. Thus, the bulk of their sites are classified as ES, making it the main topic. Late in the manuscript they note that " The distribution and deposition of deep-sea diatom-bearing sediments will be examined in a future article and compared with the results of this study". I'm glad to see this, but it should be acknowledged up front.  Furthermore, the database is heavily weighted toward outcrops in Europe and Eurasia, with exceedingly few sites in the Southern Hemisphere and only a few in North America, central Asia and none in Africa, India, Australia or Antarctica. This is, admittedly, an inherent and unavoidable weakness in an attempt at a global assessment. Compiling global literature such as this as a challenge, rife with pitfalls, but the authors do an admirable job of seeking the big picture and to their credit, they take a conservative approach. The database excludes famous Paleogene diatomites such as Oamaru diatomite (New Zealand). Without specifically naming it, they do explain why sites like Oamaru are excluded from their database (the outcrop is no longer accessible), but Oamaru is so well-documented that I'd think it should be included.

What this reviewer finds lacking is a more explicit definition of shallow versus open ocean marine diatomites. Epicontinental seas can be extensive during times of elevated sea levels across low-lying continental masses, but what about along continental margins, which may include expansive continental shelf areas, which are by definition shallow marine? Inferences distinguishing shallow water (say, under ~100m paleo water depth, from deeper paleodepths of continental slopes and rises, which would constitute true OOS, can be drawn from analysis of the diatom assemblages. For example, the occurrence of benthic, notably epipsammic, and neritic resting spore-forming taxa would go far in establishing coastal marine (continental shelf) from true open ocean settings, but this approach is not discussed in this paper.

Many Eocene marine diatomites recovered from drilling and coring along continental margins and on submerged/subsided microcontinental masses represent deposition in shallow (or relatively shallow) marginal marine settings that may not be significantly influenced by open ocean currents, especially in narrow basins with restricted flow, such as that recovered by IODP Expedition 396 and earlier drilling in the Norwegian Sea (see Berndt et al., 2023; Planke et al., 2023; Schrader and Fenner, 1976).  The cruise report from Exp. 396 notes the ecological similarity of the diatom assemblages to those of the epicontinental Russian Platform sites they include in their database, and, to a somewhat lesser extent, the Fur Formation in Denmark. These diatomites differ markedly from true

deep ocean sites. Consequently, most Vøring Plateau sites should be classified as ES deposits, although recovered from an offshore continental margin that lies in relatively deep water today. Such acknowledgement or, better still, analysis, would add length to the paper, but would greatly strengthen its impact.

The authors prominently note an ES diatomite gap between 46 and 43 Ma in epicontinental seas, and an onset of diatomite accumulation in the open ocean beginning at 43.5 Ma. Some pure, likely shallow water diatomites recently recovered from the Norwegian Sea (IODP Exp. 396, sites U1571 and U1572) are dated slightly older than this noted gap (mid-Lutecian, between ~44 and ~43.5 Ma, based predominantly on dinoflagellate biostratigraphy). Some of these diatomites may be classified as open water, but others likely reflect deposition in shallow water. Currently lacking magnetostratigraphy, we cannot be certain if these fall within the lower part of Chron C20n or the upper part of Chron C20r. If all IODP 396 and DSDP Leg 38 Paleogene diatomites are classified as OOS, the authors may be correct that middle Eocene open ocean diatomite accumulation recommenced ~43.5 Ma. But if they are classified as ES then their "gap" may need to be reduced.

The freshwater diatomite compilation does not seem to add much to the paper in terms of climatic controls, other than to note that volcanic input likely supports freshwater diatomite production and preservation. [A correlation between glassy ashes and preserved diatoms has been noted for decades (e.g., Blake, 1903; van Vleck Anderson, 1933; Bramlette, 1946).]

The introductory sections are useful and provide context, but include some discussion that is not directly supportive of the paper's theme. There is some ancillary yet still important commentary in acknowledging the potential limits of their dataset, including pointedly noting the small number of specialists working to generate high quality data on diatomaceous deposits. It is absolutely true that there is a plethora of diatom-rich sediments that have never been analyzed in any systematic way by diatom specialists, in large measure due to the lack of trained personnel, which goes back to the precipitous decline in funding for micropaleontological studies over the last several decades. Although often perceived as an "old-fashioned" science, these assemblages carry with them a vast amount of information regarding biostratigraphic age and paleoenvironments that could never be gleaned from single geochemical proxies. The limitations and uncertainties associated with diatom analysis have always been set by the low numbers of trained diatom workers. Many of us have been aging out, and the lack of career positions and resources is the sole barrier preventing growth among the next generation. I'm very pleased to see Cécile Figus and a few other young scientists addressing this problem. She and others will ensure continued important diatom science into the future. I hope hiring and funding sources will support a bright future for a new generation of fossil diatom scientists.

**Technical Comments:**

The paper is well-written with clear language and proper usage, with one potential exception. Although technically not incorrect, and admittedly a pet peeve of mine, frequent use of "*while*" (which implies the time domain: "during") would be better written as "*although*" (which is used to draw a contrast). There are one or two typos that I'm sure will be discovered in revision.

They note data availability on Zendos, but given that it's a small table, it should also be posted in Supplementary Materials. Furthermore, readers would really appreciate maps with interactive links to the data file, so we can easily identify each site locality.

**Concluding Remark:**
In conclusion, this is a very good paper, well researched and well worth publishing, though points noted above should be clarified and the manuscript tightened up. It is of appropriate scientific significance, scientific quality and presentation quality to justify publication as an *EGUSphere* contribution. In revision it should be made clearer that the paper is largely a compilation of available literature on Paleogene epicontinental diatomites, with some observations on similar-aged freshwater and open ocean deposits, and that the database is entirely focusing on terrestrially exposed materials, not on offshore coring. Supplementary materials should include the database linked to maps.

**References cited:**

Berndt et al., 2023. Shallow-water hydrothermal venting linked to the Palaeocene–Eocene Thermal Maximum. Nature Geoscience, 16(9), pp.803-809.

Blake, W. P. (1903). Arizona diatomite. *Transactions of the Wisconsin Academy of Sciences, Arts, and Letters*, *14*, 107.

Bramlette, M. N. (1946). *The Monterey Formation of California and the origin of its siliceous rocks* (Vol. 212). US Government Printing Office.

Planke, S., Berndt, C., Alvarez Zarikian, C.A., and the Expedition 396 Scientists, 2023. Mid-Norwegian Margin Magmatism and Paleoclimate Implications. Proceedings of the International Ocean Discovery Program, 396: College Station, TX (International Ocean Discovery Program). **https://doi.org/10.14379/iodp.proc.396.2023**

Schrader, H.J. and Fenner, J., 1976. Norwegian Sea diatom biostratigraphy and taxonomy. In, Talwani, M., Udintsev, G. et al., Init. Rep. DSDP, 38, pp.921-1099.

van Vleck Anderson, Robert. "The diatomaceous and fish-bearing Beida stage of Algeria." *The Journal of Geology* 41.7 (1933): 673-698.

---

## Author Response (AR1)

INSTITUTE OF MARINE
AND ENVIRONMENTAL SCIENCES
UNIVERSITY OF SZCZECIN

Biogeosciences Editorial Board
Copernicus Publications
Bahnhofsallee 1e
37081 Göttingen
Germany

Szczecin, October 2, 2024

Dear Dr. Voelker,

We thank you for the opportunity to revise our manuscript entitled 'Climatic and tectonic controls on shallow marine and freshwater diatomite deposition through the Palaeogene'. We have added minor corrections to typos, marked-up in the manuscript, and changed the affiliation of one of the co-authors from 'Petroleum Higher School, Almetyevsk, 423450, Russia' to 'RUDN University, Moscow, 117198, Russia'. Please find below detailed responses to the reviewer's suggestions.

Yours sincerely,
Cécile Figus, on behalf of all Co-Authors

**Responses to Reed Scherer, Northern Illinois University:**

General Comments:

Diatoms play a key role in global biogeochemical cycling, contributing to atmospheric oxygen levels and to $CO_2$ regulation through sequestration in geological deposits. Diatoms require high nutrients to bloom and accumulate, so intervals of diatom proliferation on and near continental masses typically correlate with times of rapid continental weathering and delivery to non-marine and marginal marine waterways; the latter including both epicontinental seas and along continental shelves, especially near river mouths that carry a high dissolved nutrient load but low sediment load. By contrast, deep-sea diatomite accumulation is typically dominated or augmented by upwelling of nutrients at oceanic divergence zones, driven by broad ocean circulation patterns. These may be along continental margins (e.g, California coast), or in open ocean settings (e.g., the Southern Ocean).

Figus et al. have analyzed occurrences of Paleogene diatomites around the world, based on literature surveys, seeking patterns that may relate to global climate changes in the Hothouse world. The analysis they present is based on literature surveys of terrestrial exposures. Their paper is an ambitious compilation that provides an important window into a critical interval in Cenozoic history. Their chief aim is to evaluate the hypothesis that Paleogene diatomite accumulation was driven predominantly by climate change. One notable new observation is a possible "diatomite gap" between ~46 and ~43 Ma.

There are, of course, biases and limitations of the stratigraphic record, which they acknowledge. The greatest limitation is access to material for study. On the continents, marine diatomites will only accumulate during times and in places affected by of high sea level. Access these strata is limited because they may have become deeply buried with little or no exposure, or, once exposed, these deposits are highly susceptible to erosion. Consequently, terrestrially-based Paleogene diatomites are very rare. Access to appropriate deep-sea deposits is even more challenging. Most known oceanic Paleogene diatomites were recovered by ocean drilling around the world and are not part of this paper. Some Paleogene open ocean diatomaceous deposits are found on land in some coastal areas, having been tectonically uplifted. Paleogene deep-sea sediments, accessible only through ocean drilling, may be deeply buried, increasing their susceptibility to diagenesis and also making them less accessible through high quality drilling. Moreover, Paleogene diatomites that would have accumulated in the deep sea across global oceans have already been subducted and destroyed, thus very severely limiting assessment of global Paleogene deep sea accumulation.

Specific Comments:

The authors limit their analysis to sediments both exposed on land and technically defined as diatomite, excluding diatom-bearing muds. They distinguish three categories of Paleogene diatomite: freshwater diatomites, epicontinental sea deposits (ES) and open ocean settings (OOS). Their database lists 102 terrestrially exposed diatomites used in the analysis. Of these only 17 are classified as OOS and 20 as freshwater. Thus, the bulk of their sites are classified as ES, making it the main topic. Late in the manuscript they note that " The distribution and deposition of deep-sea diatom-bearing sediments will be examined in a future article and compared with the results of this study". I'm glad to see this, but it should be acknowledged up front.

We now introduce this idea earlier in the paper, in section 2.2 Geographic distribution.

Furthermore, the database is heavily weighted toward outcrops in Europe and Eurasia, with exceedingly few sites in the Southern Hemisphere and only a few in North America, central Asia and none in Africa, India, Australia or Antarctica. This is, admittedly, an inherent and unavoidable weakness in an attempt at a global assessment. Compiling global literature such as this as a challenge, rife with pitfalls, but the authors do an admirable job of seeking the big picture and to their credit, they take a conservative approach.

We have added a part to section 2.2 to cover this problem: 'Furthermore, there appears to be an imbalance in the study of diatomites between the northern and southern hemispheres, with a greater concentration of research on the northern hemisphere. However, statistical treatment of the compilation shows that there is in fact a higher concentration of diatomites in the northern hemisphere.'

The database excludes famous Paleogene diatomites such as Oamaru diatomite (New Zealand). Without specifically naming it, they do explain why sites like Oamaru are excluded from their database (the outcrop is no longer accessible), but Oamaru is so well-documented that I'd think it should be included.

Lithologically, the 'Oamaru Diatomite' is not a diatomite; the name is misleading. It is briefly discussed in Witkowski et al (2017), and discussed at length in Edwards (1991). For this reason, we decided to keep this site excluded from the compilation.

What this reviewer finds lacking is a more explicit definition of shallow versus open ocean marine diatomites. Epicontinental seas can be extensive during times of elevated sea levels across low-lying continental masses, but what about along continental margins, which may include expansive continental shelf areas, which are by definition shallow marine? Inferences distinguishing shallow water (say, under ~100m paleo water depth, from deeper paleodepths of continental slopes and rises, which would constitute true OOS, can be drawn from analysis of the diatom assemblages. For example, the occurrence of benthic, notably epipsammic, and neritic resting spore-forming taxa would go far in establishing coastal marine (continental shelf) from true open ocean settings, but this approach is not discussed in this paper.

Our study does not oppose the shallow to the open ocean, but the shallow to the deep-sea. In order to clarify this point and why we have considered the continental slope but not the continental rise (both of which are part of the continental margin), we have added a definition of shallow vs. deep-sea environments. This development gave us the opportunity to introduce the future deep-sea paper earlier in the manuscript, as you requested in one of your comments. As for the analysis of assemblages, this would be an interesting perspective for further study, but this is not part of our current study. Furthermore, diatom assemblage analysis may be misleading in that offshore currents can concentrate neritic taxa in open ocean settings, thus suggesting a depositional environment that does not reflect the actual depositional setting. This issue has been discussed in detail in the various Blake Nose papers.

[Figure]

Many Eocene marine diatomites recovered from drilling and coring along continental margins and on submerged/subsided microcontinental masses represent deposition in shallow (or relatively shallow) marginal marine settings that may not be significantly influenced by open ocean currents, especially in narrow basins with restricted flow, such as that recovered by IODP Expedition 396 and earlier drilling in the Norwegian Sea (see Berndt et al., 2023; Planke et al., 2023; Schrader and Fenner, 1976). The cruise report from Exp. 396 notes the ecological similarity of the diatom assemblages to those of the epicontinental Russian Platform sites they include in their database, and, to a somewhat lesser extent, the Fur Formation in Denmark. These diatomites differ markedly from true deep ocean sites. Consequently, most Vøring Plateau sites should be classified as ES deposits, although recovered from an offshore continental margin that lies in relatively deep water today. Such acknowledgement or, better still, analysis, would add length to the paper, but would greatly strengthen its impact.

At the time of preparing the compilation, not enough data were available on IODP Expedition 396 to include it in the compilation. However, thanks to the new age control details you have recently communicated to us, we have included these sites in the compilation as ES deposits, like the other sites in Northern Europe. The database in Supplementary Materials/on Zenodo, the figures and the text have been updated.

The authors prominently note an ES diatomite gap between 46 and 43 Ma in epicontinental seas, and an onset of diatomite accumulation in the open ocean beginning at 43.5 Ma. Some pure, likely shallow water diatomites recently recovered from the Norwegian Sea (IODP Exp. 396, sites U1571 and U1572) are dated slightly older than this noted gap (midLutecian, between ~44 and ~43.5 Ma, based predominantly on dinoflagellate biostratigraphy). Some of these diatomites may be classified as open water, but others likely reflect deposition in shallow water. Currently lacking magnetostratigraphy, we cannot be certain if these fall within the lower part of Chron C20n or the upper part of Chron C20r. If all IODP 396 and DSDP Leg 38 Paleogene diatomites are classified as OOS, the authors may be correct that middle Eocene open ocean diatomite accumulation recommenced ~43.5 Ma. But if they are classified as ES then their "gap" may need to be reduced.

After including these data in our study, we updated the gap to ~44-46 Ma.

The freshwater diatomite compilation does not seem to add much to the paper in terms of climatic controls, other than to note that volcanic input likely supports freshwater diatomite production and preservation. [A correlation between glassy ashes and preserved diatoms has been noted for decades (e.g., Blake, 1903; van Vleck Anderson, 1933; Bramlette, 1946).]

The freshwater sites are too few to provide any new information, but we believe they should be retained to ensure the completeness of the compilation.

The introductory sections are useful and provide context, but include some discussion that is not directly supportive of the paper's theme. There is some ancillary yet still important

commentary in acknowledging the potential limits of their dataset, including pointedly noting the small number of specialists working to generate high quality data on diatomaceous deposits. It is absolutely true that there is a plethora of diatom-rich sediments that have never been analyzed in any systematic way by diatom specialists, in large measure due to the lack of trained personnel, which goes back to the precipitous decline in funding for micropaleontological studies over the last several decades. Although often perceived as an "old-fashioned" science, these assemblages carry with them a vast amount of information regarding biostratigraphic age and paleoenvironments that could never be gleaned from single geochemical proxies. The limitations and uncertainties associated with diatom analysis have always been set by the low numbers of trained diatom workers. Many of us have been aging out, and the lack of career positions and resources is the sole barrier preventing growth among the next generation. I'm very pleased to see Cécile Figus and a few other young scientists addressing this problem. She and others will ensure continued important diatom science into the future. I hope hiring and funding sources will support a bright future for a new generation of fossil diatom scientists.

Technical Comments:

The paper is well-written with clear language and proper usage, with one potential exception. Although technically not incorrect, and admittedly a pet peeve of mine, frequent use of "while" (which implies the time domain: "during") would be better written as "although" (which is used to draw a contrast).

This has been corrected.

There are one or two typos that I'm sure will be discovered in revision.

This has been corrected.

They note data availability on Zendos, but given that it's a small table, it should also be posted in Supplementary Materials.

The table is now also available in the Supplementary Materials.

Furthermore, readers would really appreciate maps with interactive links to the data file, so we can easily identify each site locality.

Unfortunately, this was not possible to achieve with the means at our disposal.

Concluding Remark:

In conclusion, this is a very good paper, well researched and well worth publishing, though points noted above should be clarified and the manuscript tightened up. It is of appropriate scientific significance, scientific quality and presentation quality to justify publication as an EGUSphere contribution. In revision it should be made clearer that the paper is largely a compilation of available literature on Paleogene epicontinental diatomites, with some observations on similar-aged freshwater and open ocean deposits, and that the database is entirely focusing on terrestrially exposed materials, not on offshore coring.

[Figure]

We have added information to the text to make this point clearer.

Supplementary materials should include the database linked to maps.

References cited:

Berndt et al., 2023. Shallow-water hydrothermal venting linked to the Palaeocene–Eocene Thermal Maximum. Nature Geoscience, 16(9), pp.803-809.

Blake, W. P. (1903). Arizona diatomite. Transactions of the Wisconsin Academy of Sciences, Arts, and Letters, 14, 107.

Bramlette, M. N. (1946). The Monterey Formation of California and the origin of its siliceous rocks (Vol. 212). US Government Printing Office.

Planke, S., Berndt, C., Alvarez Zarikian, C.A., and the Expedition 396 Scientists, 2023. Mid-Norwegian Margin Magmatism and Paleoclimate Implications. Proceedings of the

International Ocean Discovery Program, 396: College Station, TX (International Ocean Discovery Program). https://doi.org/10.14379/iodp.proc.396.2023

Schrader, H.J. and Fenner, J., 1976. Norwegian Sea diatom biostratigraphy and taxonomy. In, Talwani, M., Udintsev, G. et al., Init. Rep. DSDP, 38, pp.921-1099.

van Vleck Anderson, Robert. "The diatomaceous and fish-bearing Beida stage of Algeria." The Journal of Geology 41.7 (1933): 673-698.

[Figure]

**Responses to Anonymous Reviewer 2:**

The manuscript is focusing on diatoms, which are crucial to the carbon and silicon cycles. Diatoms form diatomite in shallow marine and freshwater environments, serving as key indicators of past climatic and environmental conditions. This review compiles diatomite occurrences from the Palaeogene period (66 to 23 million years ago) to understand the impact of climatic, tectonic, and oceanographic changes on diatom production. The compilation is analysed with a focus on the influence of palaeogeographic and palaeoceanographic shifts on diatomite deposition, particularly during the Early Eocene Climatic Optimum and towards the upper Paleogene.

The paper address relevant scientific questions within the scope of the journal. It presents a compilation of the existing data and aims to investigate a link between the diatom production and the climatic evolution of the Paleogene. I don't follow the details beyond the site selection, it may be due to linguistic issues with the section (lines 84-92). Perhaps the map on Fig. 1 could also show how many sites exists, which do not fulfil the criteria? The most novel observation of this work is a "period without diatomite occurrences" between ~46 and ~43 Ma.

It would not be possible to show all the sites that do not meet the criteria, as some of them have been rejected due to a lack of precise location. In addition, it would make the map too heavy and unclear.

Are the abbreviations of the epicontinental seas (abbreviated as ES) and open ocean settings (OOS) really needed? I suggest skipping it. If authors want to keep it, then I think that Freshwater setting should also be abbreviated.

Epicontinental seas and open ocean settings are abbreviated to make the paper less redundant and less lengthy, as these two names take up a lot of space, considering the number of times they are written in the manuscript, unlike freshwater, which takes up less space and is repeated less often.

But what is more important, I find these "categories/setting types" explained in a bit to superficial way.

We have developed the explanation of the categories a little further in section 2.2.

Regarding the chronology: why GTS20212 and not GTS2020? What about age of shallow marine diatomites? It looks like this section is only about freshwater diatoms.

The studies compiled for comparison with our results were mainly calibrated on GTS2012. The average difference with GTS2020 is only 0.1 Myr over the Palaeogene (the largest difference being 0.5 Myr, which is less than the 1 Myr binning we use).

Statistical treatment: it seems like the occurrence s are divide into clusters regarding the age (1 million year time span) and geographically. But what are these geographical clusters based on? Present-day distribution? Is a number of data points per cluster important here?

The names of the geographical clusters refer to the present-day names of regions, but the clusters reflect the 'regions' concerned during the Palaeogene.

Most of the study does not involve clustering, but the clustering applied here is important, as it allows us to see whether the global curves (ES and OOS) are in fact global, or whether they only reflect local processes.

Some of the parts of discussion fits better in the introduction

We would need more details to know which parts are concerned.

Figure 1 and 4: The points on maps are difficult to distinguish.

We have now increased the size of the points on these figures.

It may be confusing when merging names of the events and ages in the same sentence, e.g. line 192: "A comparison between ES deposits and climate proxies between the end of the EECO and ~43 Ma" I think that it could be: " Between ~49 (the end of the EECO) and ~43 Ma we observe a clear correlation between the epicontinental sea deposits and the climate proxy records".

We have made changes to improve that.

1. Are substantial conclusions reached? yes
2. Are the scientific methods and assumptions valid and clearly outlined? Almost, see my comments above.
3. Are the results sufficient to support the interpretations and conclusions? yes
4. Is the description of experiments and calculations sufficiently complete and precise to allow their reproduction by fellow scientists (traceability of results)? Yes
5. Do the authors give proper credit to related work and clearly indicate their own new/original contribution? Partially. Due to not sufficient references, it is sometimes difficult to differentiate what is an original observation and what is already observed by others (see my comment below regarding references)
6. Does the title clearly reflect the contents of the paper? Yes
7. Does the abstract provide a concise and complete summary? Yes
8. Is the overall presentation well-structured and clear? I would like to see some of the climatic events mentioned in the discussion (e.g. EOT, Latest Danian Event), introduced a bit better – what is their signature and significance for the global climate?

We have now developed this point when introducing the most important hyperthermals in the discussion.

9. Is the language fluent and precise? There are many very long sentences, so sentence structure could be improved. Also some other linguistic changes would be beneficial, such as: "Figure 2E shows…" it would sound better "Our data (Fig. 2E) show that…" ; " the middle Eocene was eroded in several locations" I assume it means that "deposits of Eocene age were eroded"? There are many examples of that kind in the text, and it will need to be corrected.

This has been corrected.

[Figure]

10. Are mathematical formulae, symbols, abbreviations, and units correctly defined and used? Not all (e.g. H4SIO4, CO2), please check through text. I think also that the number of abbreviation is unnecessary high (e.g. why introduce the abbreviation "DDE", if it is used only once?) .

We have replaced 'silicic acid' (H4SiO4) by 'dissolved silica' in the text, to make it easier for readers to understand. We do not believe that CO2 needs to be explained.
Unnecessary abbreviations have been removed.

11. Should any parts of the paper (text, formulae, figures, tables) be clarified, reduced, combined, or eliminated? no
12. Are the number and quality of references appropriate? Almost, I think that some key papers for some of the paleoclimatic events would be great to add (see Nr. 8) , e.g Hutchinson et al, 2021 for the EOT, etc…). Many statementsm such as :" corresponding to a peak in diatomite deposition in northern Europe (Fig. 3)" should be accompanied by a proper reference.

We have added the papers relating to climatic events. However, certain parts of the manuscript, such as the one cited as an example in this comment, cannot be accompanied by a reference, as they form part of our own study.

13. Is the amount and quality of supplementary material appropriate? Yes
The language is OK, but some polishing will be needed for the final publication.

The text has been revised to improve this.

Minor comments:
Line 28 – deep sea drilling campaigns did not generate the records; the records were generated on the sediment cores from the drilling campaigns

This has been corrected.

Line 37 – I suggest using/adding a more relevant references, e.g. Westerhold et al. 2020, Zachos et al. 2001.

These references have been added.

Line 57 – this record is far from global

Our compilation includes diatomites from North and South America, Africa, Europe, Russia, Asia and New Zealand. We believe that the term 'global' is appropriate.

Line 143 – consuming?

Silicate weathering consumes CO2 through the reaction of silicate rocks with water and CO2, releasing dissolved silica through chemical weathering.